# The TUDOR domain of SMN is an H3K79$^{me1}$ histone mark reader

Olivier Binda[1,2] , Aimé Boris Kimenyi Ishimwe[1], Maxime Galloy[3], Karine Jacquet[1], Armelle Corpet[1] , Amélie Fradet-Turcotte[3] , Jocelyn Côté[2], Patrick Lomonte[1] 

Spinal muscular atrophy is the leading genetic cause of infant mortality and results from depleted levels of functional survival of motor neuron (SMN) protein by either deletion or mutation of the *SMN1* gene. SMN is characterized by a central TUDOR domain, which mediates the association of SMN with arginine methylated (R$^{me}$) partners, such as coilin, fibrillarin, and RNA pol II (RNA polymerase II). Herein, we biochemically demonstrate that SMN also associates with histone H3 monomethylated on lysine 79 (H3K79$^{me1}$), defining SMN as not only the first protein known to associate with the H3K79$^{me1}$ histone modification but also the first histone mark reader to recognize both methylated arginine and lysine residues. Mutational analyzes provide evidence that SMN$_{TUDOR}$ associates with H3 via an aromatic cage. Importantly, most SMN$_{TUDOR}$ mutants found in spinal muscular atrophy patients fail to associate with H3K79$^{me1}$.

## Introduction

Loss of survival of motor neuron 1 (*SMN1*) gene was found in 1995 as responsible for the monogenic pathology spinal muscular atrophy (SMA) (Lefebvre et al, 1995). SMN, the protein, also known as GEMIN1, is principally recognized as a marker of membraneless nuclear structures called Cajal bodies, first identified in neuronal tissues by Santiago Ramón y Cajal (Gall, 2000) and recently found to phase separate (Binda et al, 2022). SMN's most documented cellular function is to assemble small nuclear ribonucleoproteines and thus regulate RNA metabolism and splicing, but it has other cellular roles (reviewed in the study by Beattie & Kolb [2018]; Nussbacher et al [2019]). Essentially, SMN cellular functions are centered on its TUDOR domain, which coordinates protein–protein interactions with arginine methylated (R$^{me}$) proteins, such as coilin (Boisvert et al, 2002; Hebert et al, 2002), RNA pol II (Zhao et al, 2016), and other R$^{me}$GG motif proteins (Thandapani et al, 2013, 2015; Binda et al,

2022). The TUDOR domain is found in several proteins in humans and part of a wide family called readers (including CHROMO, PHD, *etc.*) that associate with methylated histones and otherwise post-translationally modified proteins in general (see review by Musselman et al [2012]).

In humans, there are two copies of the gene that encodes SMN, namely, *SMN1* and *SMN2*. The *SMN1* and *SMN2* genes are nearly identical in sequence, differing in few nucleotides, but the *SMN2* duplication contains, among other changes, a pyrimidine transition (cytosine to thymine) in exon 7 that introduces an exonic splicing silencer element, leading to the prevailing exclusion of exon 7 (SMN$_{\Delta 7}$) and subsequent production of a truncated, unstable, and rapidly degraded protein (Lorson & Androphy, 2000). Thus, the loss of *SMN1* gene in most SMA patients leads to low levels of SMN protein, but approximately 10% of SMA cases harbor mutations. Interestingly, most SMN mutations congregate either within the carboxy terminal oligomerization domain or within the central TUDOR domain (reviewed Lomonte et al [2020]), suggesting an important role for the TUDOR domain and the capacity of SMN to oligomerize in the maintenance of motor neuron homeostasis. The TUDOR domain is part of a large family of histone- and nonhistone-recognizing modules that associate with post-translationally modified or unmodified proteins (Gayatri & Bedford, 2014; Weaver et al, 2018). Previous work from our team suggests that SMN associates with histone H3 and localizes with damaged centromeres in a DOT1L methyltransferase-dependent manner (Sabra et al, 2013). In humans, there are over 50 lysine methyl-transferases that modify histone and nonhistone proteins (Husmann & Gozani, 2019). Most lysine methylation sites on histones are located on the unstructured amino terminal tail (e.g., trimethylation of histone H3 on lysine 4 [H3K4$^{me3}$] or lysine 9 [H3K9$^{me3}$]), but lysine methylation occurs also in the core part of H3, such as on lysine 79 (H3K79). There is only one generally recognized lysine methyltransferase that catalyzes methylation of H3 on lysine 79, DOT1L. Although DOT1L is often depicted as the only histone H3 lysine 79 (H3K79) methyltransferase (Feng et al, 2002; Wood et al,

[1]Université Claude Bernard Lyon 1, CNRS UMR 5261, INSERM U1315, LabEx DEV2CAN, Institut NeuroMyoGène-Pathophysiology and Genetics of Neuron and Muscle (INMG-PGNM), Team Chromatin Dynamics, Nuclear Domains, Virus, Lyon, France   [2]University of Ottawa, Faculty of Medicine, Department of Cellular and Molecular Medicine, Ontario, Canada   [3]Université Laval Cancer Research Center, Université Laval, Québec, Canada; Department of Molecular Biology, Medical Biochemistry and Pathology, Université Laval, Québec, Canada; and Oncology Division, Centre Hospitalier Universitaire (CHU) de Québec-Université Laval Research Center, Québec, Canada

Correspondence: olivier.binda@mail.mcgill.ca; patrick.lomonte@univ-lyon1.fr

2018), $Dot1l^{-/-}$ knockout cells retain H3K79$^{me2}$ albeit at extremely low levels (0.5% in $Dot1l^{-/-}$ versus 3.3% H3K79$^{me2}$ in WT cells) (Jones et al, 2008), suggesting that there may be other methyltransferase(s) capable of modifying H3K79. There are indeed a few studies suggesting that NSD1 and NSD2 methyltransferases could mono- and dimethylate H3K79 (Morishita et al, 2014; Park et al, 2015).

Histones and histone post-translational modifications (histone marks) are central to chromatin signalling pathways. Essentially, genomic DNA is wrapped around small basic proteins called histones to form nucleosomes, a repetitive unit constituting the chromatin framework, which regulates access to genetic information. Generally, histone modifications such as H3K4$^{me3}$, H3K9$^{me1}$, and H3K79$^{me1}$ mark the chromatin for access to the genetic information, whereas modifications such as H3K9$^{me3}$ and H3K27$^{me3}$ mark restrict access to the genetic information (Barski et al, 2007). Regarding H3K79$^{me1}$, the mark correlates with alternative splicing patterns between cell lines (Shindo et al, 2013), in agreement with the positioning of H3K79$^{me1}$-marked nucleosomes on exons (Andersson et al, 2009). More precisely, the H3K79$^{me1}$ and H3K79$^{me2}$ marks are found at alternative 3' and 5' splice sites (Zhou et al, 2012). The H3K79$^{me1}$ mark is the most prominent state of methylation on H3K79 in mouse ES cells (Jones et al, 2008). Notably, the mouse model of $Dot1l^{-/-}$ is embryonic lethal, although $Dot1l^{-/-}$-derived cells harbor alternative lengthening of telomere phenotype (Jones et al, 2008). Moreover, DOT1L plays an important role in neuronal development (Franz et al, 2019).

Herein, using biochemical approaches, purified recombinant proteins, and recombinant nucleosomes, we define SMN$_{TUDOR}$ as the first protein known to associate with H3K79$^{me1}$-modified chromatin and also the first histone mark reader capable of reading both R$^{me}$ and K$^{me}$ states. Importantly, SMA-linked SMN$_{TUDOR}$ mutations (SMN$_{ST}$) prevent SMN-H3 interactions, suggesting the involvement of chromatin signalling pathways in SMA genetic pathology.

# Results

## Purified recombinant SMN interacts directly with histone H3 in vitro

We previously reported that SMN associates with damaged centromeres in a DOT1L-dependent fashion (Sabra et al, 2013), suggesting that SMN could associate with histone H3 methylated on lysine 79. Indeed, initial investigations suggested that SMN associates with H3K79$^{me2}$ peptides (Sabra et al, 2013). Herein, we thus aimed to further characterize biochemically this potential interaction. Using recombinant human SMN purified from *Escherichia coli* by a GST (glutathione S-transferase) affinity purification scheme, GST-SMN was subjected to pulldown assays in the presence of calf thymus histones, a classic source of modified histones (Allfrey et al, 1964). In agreement with previous work (Sabra et al, 2013), GST-SMN was capable of associating directly with histone H3, whereas the GST alone control failed to associate with histones (Fig 1A). Similar experiments were conducted with the PHD domain of the H3K4$^{me3}$ reader ING3 (ING3$_{PHD}$) and a characterized aromatic cage mutant known to be unable to associate with H3 (ING3$_{W385A}$)

(Kim et al, 2016; McClurg et al, 2018), as positive and negative controls, respectively. These pulldowns were analyzed by immunoblotting against core histones and histone variants. As expected, ING3$_{PHD}$ associated with H3, whereas the aromatic cage mutant ING3$_{W385A}$ failed to do so (Fig 1B). As seen in Fig 1A, we observed that GST-SMN associates predominantly with H3 (Fig 1B). Extended immunoblots for each pulled-down protein are provided in supplementary materials (Fig S1A). These experiments validate the association of SMN with histone H3.

## An intact TUDOR domain is required for SMN to interact with H3

As SMN harbors a TUDOR domain, which is found in several other histone mark readers (Musselman et al, 2012), we then set out to define the minimal region of SMN required for H3-binding and generated a panel of truncated forms (Fig 2A). Using these, we found that deletion of either the amino terminus (SMN$_{ΔN}$) or the carboxy terminus (SMN$_{ΔC}$) affected the association with H3 minimally (Fig 2B). However, the TUDOR domain on its own (SMN$_{TUDOR}$) failed to associate with H3 when compared with the full-length form of SMN but seemed to be required for the interaction because the amino terminus (SMN$_{Nterm}$) and carboxy terminus (SMN$_{Cterm}$), which lack the TUDOR domain, only bound weakly to H3 (Fig 2B). Thus, we conclude that the TUDOR domain is required in vitro but not sufficient for SMN to associate with H3. We then extended the TUDOR domain on both sides, with actual SMN WT sequences, and found that an extension by 25 amino acid residues on its amino terminal side restored to some extent the association of SMN with H3 (Fig 2C). Extension on the carboxy terminal end of SMN$_{TUDOR}$ by 5, 10, or 15 residues did not appear to improve the association of SMN to histone H3 (Fig S1B). We conclude that SMN$_{TUDOR}$ is necessary and sufficient for association with histone H3, but requires additional residues outside the classical defined TUDOR domain.

The solution structure of SMN bound to R$^{me}$ residue displays an aromatic cage composed of W102, Y109, Y127, and Y130 (Tripsianes et al, 2011). Aromatic cages are broadly found in histone mark readers and involved in sensing methylation states (Musselman et al, 2012). To demonstrate the importance of the TUDOR domain in mediating the interaction between SMN and H3, we have mutated W102, Y109, Y127, and Y130 aromatic cage sites (SMN$_{AC}$ mutants) individually to alanines and performed pulldown assays to assess the interaction of SMN mutants with H3. Unexpectedly, unlike other readers such as ING4 (Hung et al, 2009), HP1α (Irving-Hooper & Binda, 2016), or MPP8 (Irving-Hooper & Binda, 2016), single mutations to alanine within the aromatic cage did not appear to alter SMN$_{AC}$ binding to H3 and retained the capacity to associate with H3 (Fig 3A). We thus mutated these four residues in various combinations and found that W102 and Y130 seemed to be the most important residues of the aromatic cage as all the SMN forms that had reduced binding affinity to H3 contained W102A and Y130A mutations (Fig 3B and C). Together, these results demonstrate that SMN associates directly with H3 and requires an intact TUDOR domain aromatic cage.

## SMA-linked SMN$_{TUDOR}$ mutants fail to associate with H3

In about 10% of SMA cases, *SMN1* is not deleted but mutated. These mutations aggregate mostly in the dimerization domain or within

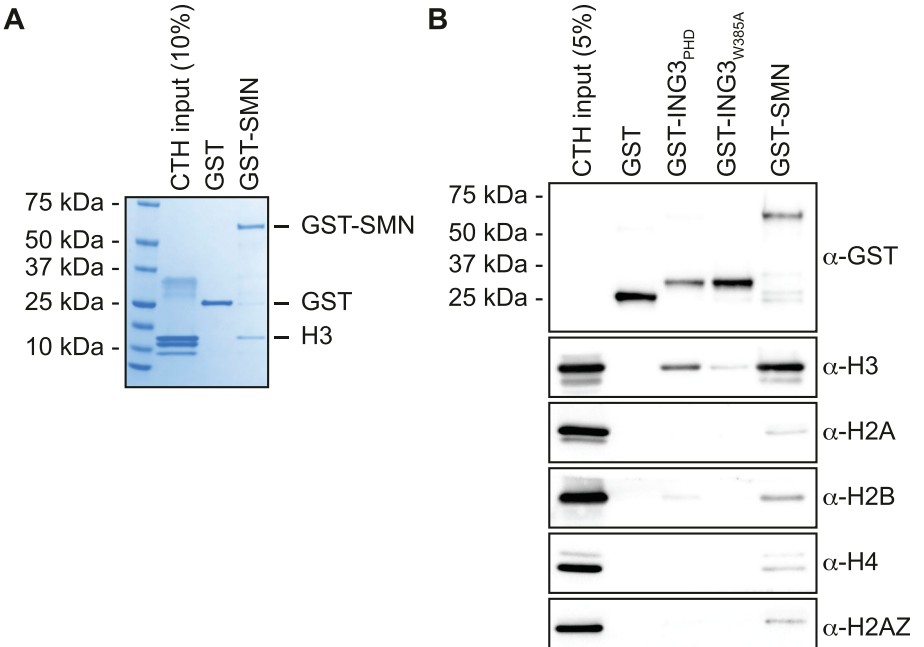

**Figure 1. SMN associates with histone H3.**
**(A)** GST alone or GST-tagged SMN was used in GST-pulldown assays in the presence of calf thymus histones. Pulldowns were analyzed by SDS–PAGE followed by Coomassie staining. **(A, B)** As in panel (A), but pulldowns were analyzed by immunoblotting using the indicated antibodies. Experiments were performed at least three times.

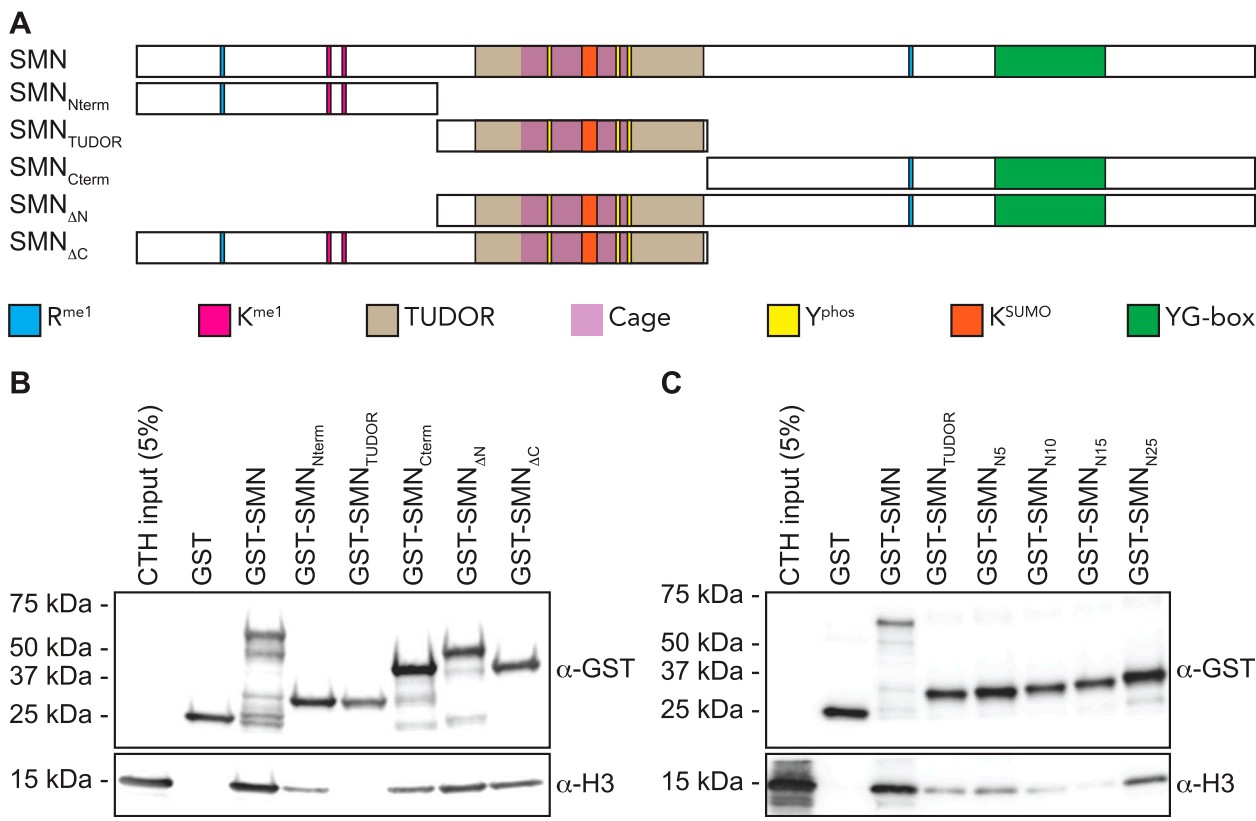

**Figure 2. The TUDOR domain is required but not sufficient for SMN to associate with H3.**
**(A)** Schematic representation of SMN truncated forms used to assess the region of SMN responsible for the association with H3. Post-translational modification (methyl–arginine [$R^{me}$], methyl–lysine [$K^{me}$], phospho–tyrosine [$Y^{phos}$], and SUMOylated lysine [$K^{SUMO}$]) sites are highlighted. The aromatic cage within the TUDOR is marked in purple, and the dimerization YG-box is highlighted in green at the carboxy terminus. **(B)** GST-pulldowns were performed in the presence of CTH and analyzed by immunoblotting with α-GST and α-H3 antibodies. **(B, C)** As in panel (B), but with amino terminal extensions on recombinant SMN$_{TUDOR}$. Experiments were performed at least three times.

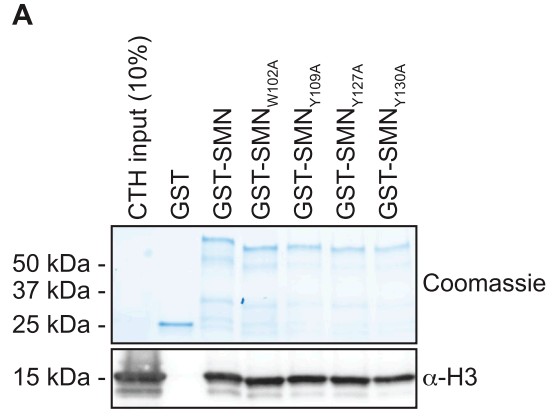

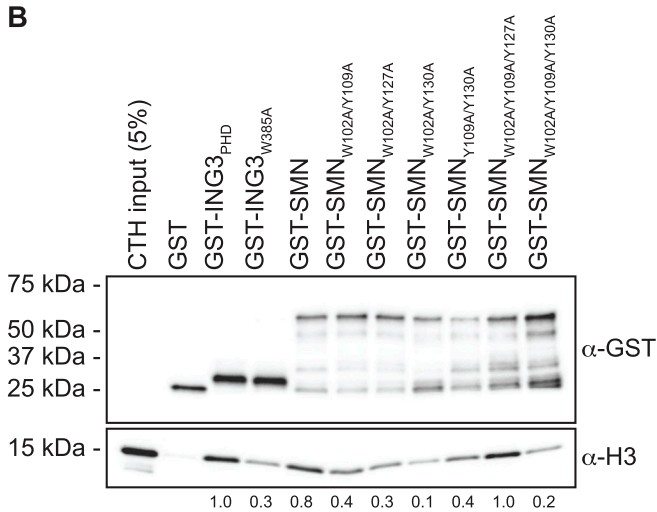

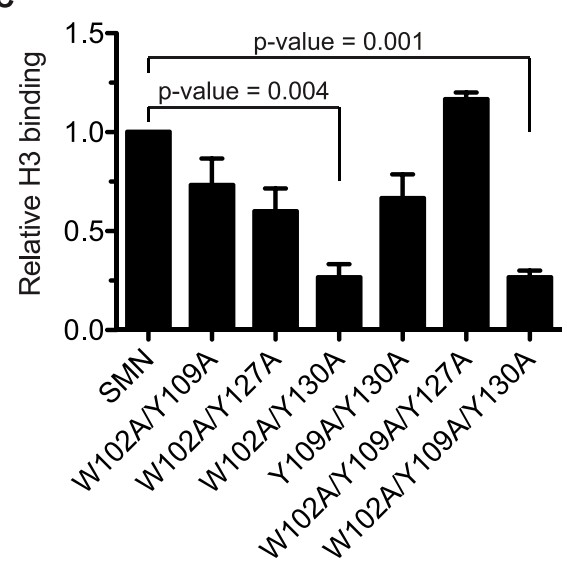

**Figure 3. The aromatic cage within SMN$_{TUDOR}$ is critical for SMN-H3.**
**(A)** GST-pulldowns were performed in the presence of CTH and analyzed by Coomassie staining (to reveal GST levels) or by immunoblotting with an α-H3 antibody. **(A, B)** As in panel (A), but GST-ING3$_{PHD}$ and ING3$_{W385A}$ were used as positive and negative controls, respectively. **(C)** Level of H3 was measured using Image Lab (Bio-Rad Laboratories) from three independent experiments.

the TUDOR domain (reviewed in the study by Lomonte et al [2020]). Given that SMN associates with H3 through an aromatic cage within its TUDOR domain (Figs 2 and 3), we investigated whether SMA-linked TUDOR mutations (SMN$_{ST}$) impact the association of SMN with histone H3. Experiments with the aromatic cage mutant SMN$_{Y109C}$ and the other SMA-linked mutant SMN$_{E134K}$ seemed to show that E134K minimally impacts the capacity of SMN to interact with H3 (Fig 4A). We thus expanded our panel to include all known SMN$_{ST}$ (Lomonte et al, 2020). Interestingly, SMN$_{ST}$ W92S, G95R, A111G, and I116F had impaired capacity to associate with histone H3, whereas V94G, Y109C, Y130C, E134K, and Q136E retained approximately WT level of binding to H3 (Fig 4B). These results suggest that SMA may arise in some cases from impaired SMN-H3 interactions or other interactions requiring an intact SMN$_{TUDOR}$.

### Defining SMN as the first H3K79$^{me1}$ reader

Previous work suggests that SMN may associate with H3 sequences surrounding the lysine 79 methylation site (Sabra et al, 2013). We thus analyzed GST-SMN pulldowns by immunoblotting against methylated H3K79 forms and found that the H3 species that associate with SMN are predominantly the H3K79$^{me1}$ and H3K79$^{me2}$ forms (Fig 5A). Although controversial, another TUDOR domain protein, 53BP1, was also reported to associate with methylated H3K79 (Huyen et al, 2004; Botuyan et al, 2006). We thus investigated how SMN associates to histones compared with 53BP1 and found that SMN bound to H3 with the H3K79$^{me2}$ mark and modestly to H4 with the H4K20$^{me2}$ modification at least as well as 53BP1 (Fig 5B). We then used the H3K4$^{me3}$ reader ING3$_{PHD}$ as a control to assess the enrichment of the H3K79 methyl marks by SMN. We found that the H3 pulled down by SMN was enriched with the H3K79$^{me1}$ mark compared with ING3$_{PHD}$ (Fig 5C), suggesting that SMN may associate with this mark (i.e., H3K79$^{me1}$). Hitherto, our results show that SMN associates with H3 harboring the H3K79$^{me1/me2}$ marks but does not demonstrate that SMN associates with the H3K79$^{me1/me2}$ marks themselves.

As mentioned above, although SMN associates with H3 methylated on lysine 79 (Figs 4 and 5), our results do not demonstrate that SMN associates directly with the H3K79 methylated marks. We thus performed pulldowns using synthetic biotinylated peptides, which are either unmodified, mono-, di-, or tri-methylated on H3K79 and recombinant SMN. While performing peptide pulldown assays, we found that SMN associates preferentially with H3K79$^{me1}$, whereas 53BP1$_{TUDOR}$ bound to the H3K79$^{me2}$ peptide (Fig 5D). We thus conclude that SMN associates directly with H3K79$^{me1}$, at least in vitro using biochemical assays.

Interestingly, the tyrosine residues (Y109, Y127, and Y130) of the aromatic cage are reported to be phosphorylated (Husedzinovic et al, 2014). Tyrosine phosphorylation within aromatic cages is hypothesized to regulate reader–mark interactions (Irving-Hooper & Binda, 2016). Conversion of Y109, Y127, or Y130 to aspartic acid, to mimic the negative charge of the phosphate moiety, appears to reduce the binding of SMN to H3K79$^{me1}$ (Fig 5E), suggesting that Y$^{phos}$ of SMN$_{TUDOR}$ could regulate the association of SMN with methylated partners, such as H3. Our in vitro biochemical characterization identifies SMN as the first known H3K79$^{me1}$ histone mark reader.

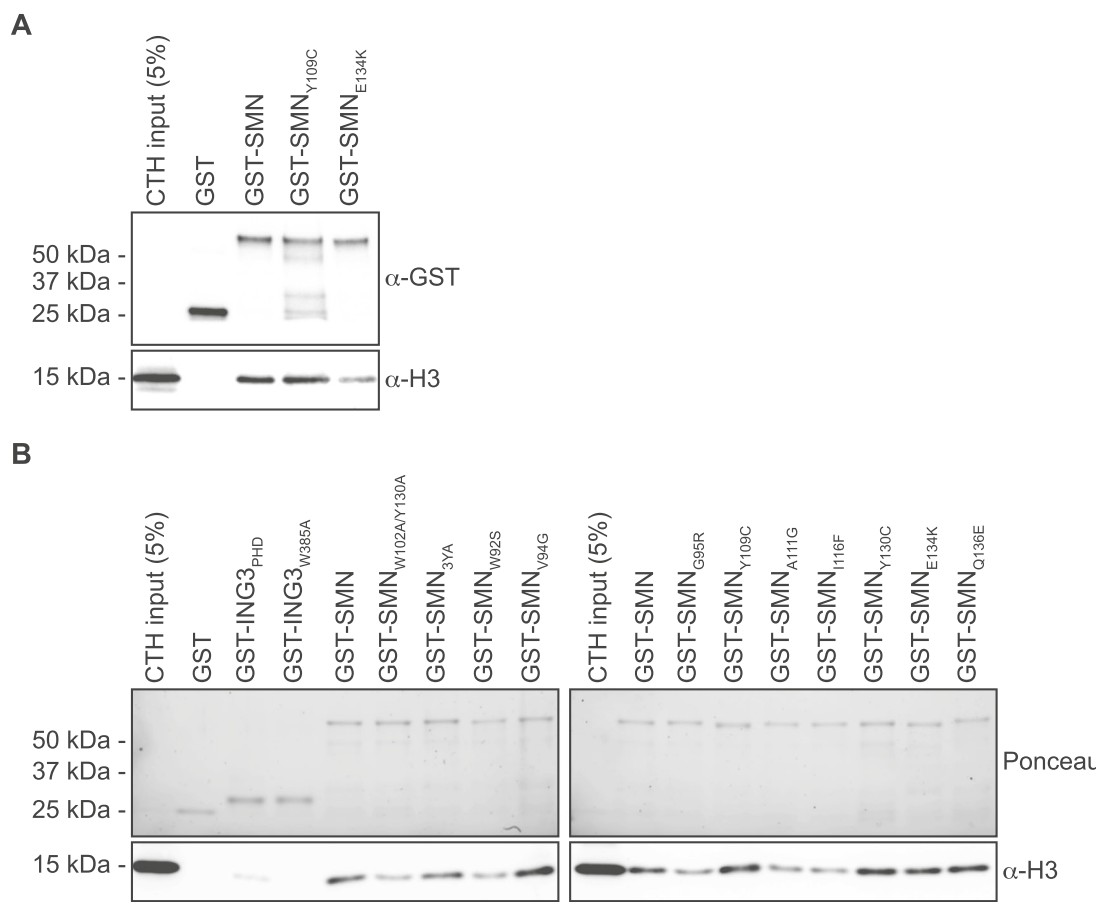

**Figure 4. SMA-linked SMN$_{TUDOR}$ mutants impact SMN interaction with H3.**
**(A)** GST-pulldowns were performed with full-length recombinant GST-SMN and SMA-linked mutants GST-SMN$_{Y109C}$ and GST-SMN$_{E134K}$ in the presence of CTH and analyzed by immunoblotting with α-GST and α-H3 antibodies. **(A, B)** As in panel (A), but with a complete panel of TUDOR mutants. The pulldowns were analyzed by immunoblotting against histone H3. Experiments were performed at least twice.

As we defined SMN as an H3K79$^{me1}$ reader, we thus then assessed the impact of SMN$_{ST}$ on binding to H3 methylated on K79. Similar to H3 (Fig 4B), W92S, Y109C, A111G, and I116F had an impact on the association of SMN with H3K79$^{me1}$, whereas V94G, G95R, Y130C, E134K, and Q136E had no apparent effect on the binding of SMN (Fig 5F).

The H3K79$^{me1}$ mark is, unlike histone modifications, such as H3K4$^{me3}$ and H3K9$^{me3}$, not found on the unstructured histone tail but within the histone H3 core region (Luger et al, 1997). Thus, the nucleosome inherent structure may impact the accessibility of H3K79$^{me1}$ to potential readers, such as SMN. To assess this possibility, we generated recombinant nucleosome core particles (rNCPs) containing histone H3 containing mutations C110A and K79C (K$_C$79) to specifically modify K$_C$79 with a monomethyl-nucleoside analog. Unmodified H3K$_C$79 (rNCP-H3K$_C$79$^{me0}$) or monomethylated H3K$_C$79 (rNCP-H3K$_C$79$^{me1}$) was used to reconstitute octamer and subsequently nucleosome assembly (Fig S2). These were used in GST-SMN pulldown experiments, which confirmed that SMN associates with H3K79$^{me1}$ within the nucleosomal context (Fig 6). Specifically, SMN associated with rNCP-H3K$_C$79$^{me1}$ but not with unmodified rNCP-H3K$_C$79$^{me0}$ form (Fig 6). Moreover, most SMA-

linked SMN$_{TUDOR}$ mutants failed to associate with H3K$_C$79$^{me1}$ nucleosomes (Fig 6). Specifically, W92S, G95R, Y109C, A111G, I116F, and Y130C mutants prevent SMN from associating with H3K$_C$79$^{me1}$ nucleosomes, whereas V94G, E134K, and Q136E had no discernable impact on the SMN-H3K79$^{me1}$ interaction (Fig 5D). Interestingly, the association of SMN and SMN$_{ST}$ with free H3, peptides, and rNCP varied slightly (Table S1). More precisely, G95R bond H3K79$^{me1}$-marked H3, but not the rNCP-H3K$_C$79$^{me1}$, whereas Y109C and Y130C bound total H3 from calf thymus but not rNCP-H3K$_C$79$^{me1}$.

# Discussion

Unlike histone tail-modified residues, such as H3K4, H3K9, or H3K27, histone H3 lysine 79 is found within the core of histone H3 with the side chain of K79 sticking out like a broken bicycle wheel spoke. Specifically, methylation of H3K79 (H3K79$^{me2}$) alters the surface of the nucleosome (Lu et al, 2008).

Although H3K79$^{me2}$ can be weakly recognized by the tandem TUDOR domain (TTD) of 53BP1 (Huyen et al, 2004), 53BP1$_{TTD}$ prefers

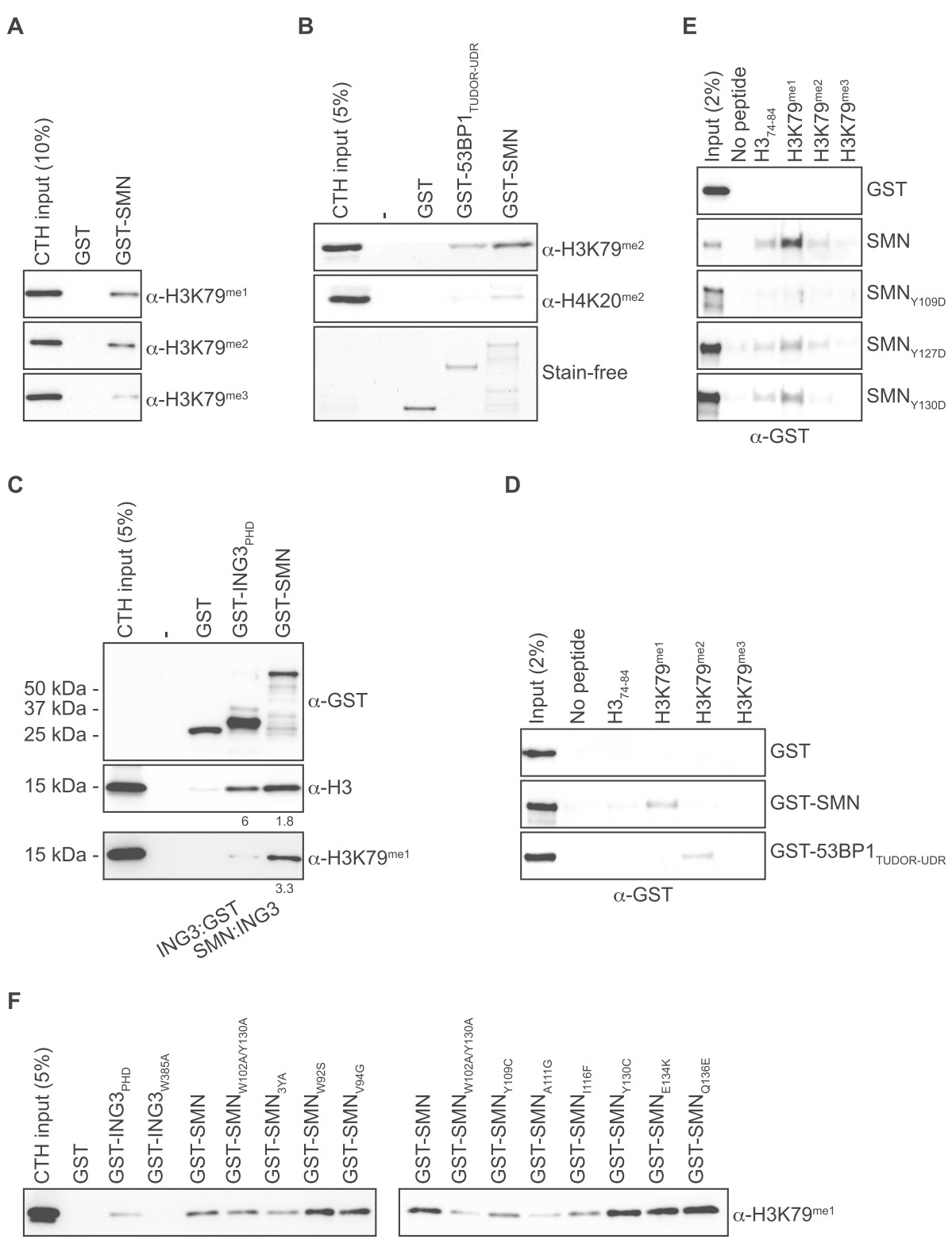

**Figure 5.  SMN is a H3K79^me1 reader.**
**(A)** GST-pulldowns with CTH were analyzed immunoblotting using α-H3K79^me1, H3K79^me2, or H3K79^me3 antibodies. **(A, B)** As in panel (A), but GST-53BP1_TUDOR-UDR was used as a positive control known to associate with H3K79^me2 (Huyen et al, 2004) and H4K20^me2 (Botuyan et al, 2006). **(A, C)** As in panel (A), but GST-ING3_PHD, a H3K4^me3 reader, was used as a negative control. Ratios of ING3: GST and SMN:ING3 signals are indicated under the immunoblots. **(D)** Biotinylated synthetic histone peptides were pulled down using streptavidin–sepharose in the presence of GST, GST-SMN, or GST-53BP1_TUDOR-UDR. Pulldowns were analyzed by immunoblotting using an α-GST antibody. **(D, E)** As in panel (D), biotin H3K79 peptides were pulled down in the presence of GST-SMN aromatic cage mutants. **(B, F)** As in Fig 4, panel (B), but the same samples were analyzed with an α-H3K79^me1 antibody. Experiments were performed at least twice.

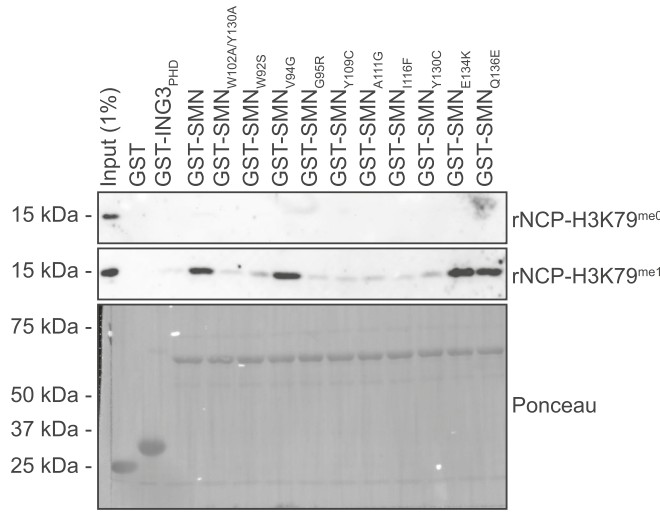

**Figure 6. SMN associates with H3K$_C$79$^{me1}$ nucleosomes.**
Recombinant nucleosomes core particles were assembled and chemically modified on lysine 79 (rNCP-H3K$_C$79$^{me0}$ and rNCP-H3K$_C$79$^{me1}$). Indicated GST-tagged proteins were pulled down using glutathione–sepharose in the presence of either rNCP and analyzed by immunoblotting using an α-H3 antibody or by Ponceau staining to detect GST-tagged proteins. Pulldowns with rNCP were performed twice.

H4K20$^{me2}$ with a K$_D$ of 20 μM compared with 2 mM for H3K79$^{me2}$ (Botuyan et al, 2006). Another study found that H3K79$^{me3}$ can be recognized in vitro by the WD40 domain of EED with low affinity (K$_D$ of > 400 μM) (PDB 3JZH [Xu et al, 2010]). Although the TUDOR domain protein fragile X mental retardation protein (FMRP) associates with H3K79$^{me2/me3}$ with a K$_D$ of 135 nM, it also binds to H3K4$^{me2/me3}$, H3K9$^{me3}$, H3K27$^{me1/me2/me3}$, and H3K36$^{me2/me3}$ (Alpatov et al, 2014). However, the FMRP family members FXR1 and FXR2 fail to associate with H3K79$^{me3}$, but rather have affinity for H4K20$^{me3}$ (Adams-Cioaba et al, 2010). Finally, there is also HDGF2$_{PWWP}$ that can recognize H3K79$^{me3}$ (Wu et al, 2011). Regardless of all these readers that can recognize H3K79$^{me2/me3}$, no readers for H3K79$^{me1}$ have been identified until now. Previous work from our laboratory described a DOT1L-dependent relocation of SMN in response to damaged centromeres that required an intact TUDOR domain (Sabra et al, 2013), suggesting that SMN$_{TUDOR}$ may interact with methylated H3K79. We herein define using multiple assays and approaches that SMN$_{TUDOR}$ associates with H3K79$^{me1}$.

Interestingly, SMA-linked SMN$_{TUDOR}$ mutants (SMN$_{ST}$) impact differently the interaction of SMN with free histone H3, K79$^{me1}$-modified H3, and H3K$_C$79$^{me1}$ rNCP (summarized in Table S1). We can only speculate on the nature of the discrepancies between the different histone contexts (i.e., peptides, free histones, rNCP), but likely oligomerization and/or nucleotide-binding properties of SMN play a role.

Although several readers recognize dual marks, such as ZMYND8$_{PHD/BD}$-H3K4$^{me1}$K14$^{ac}$ (Li et al, 2016) and SETDB1-H3K9$^{me3}$-K14$^{ac}$ (Jurkowska et al, 2017), whereas others have affinity for a handful of sites, such as PHD that recognize unmodified histone H3, H3K4$^{me}$, or H3K9$^{me}$ (Musselman & Kutateladze, 2011; Jain et al, 2020), few readers recognize both methylated arginine and lysine residues indiscriminately. For instance, SPIN1 not only recognizes

H3R8$^{me2a}$ with TUDOR-like SPIN repeat 1 and H3K4$^{me3}$ via SPIN repeat 2 (Su et al, 2014) but also H3K9$^{me3}$ and H3K4$^{me3}$ (Zhao et al, 2020). Interestingly, the overall structures of H3K4$^{me3}$R8$^{me2a}$- or H3K4$^{me3}$K9$^{me3}$-bound SPIN1 remain similar, except for a small rotation of W72 and F251 side chains in the aromatic cage of SPIN repeat 1 to accommodate either H3K9$^{me3}$ or H3R8$^{me2a}$ (Zhao et al, 2020). We conclude that SMN$_{TUDOR}$ likely conforms SPIN1 to be able to associate with H3K79$^{me1}$ and R$^{me}$GG-containing proteins (e.g., COIL, FBL).

Giving the numerous roles of SMN in cellular biology and the central part of the TUDOR domain, we can only speculate here, but SMN$_{ST}$ definitively impacts protein–protein (Binda et al, 2022) and SMN-H3 interactions (Figs 4, 5, and 6). Although SMN was previously found to associate with H3R2$^{me1}$, H3R2$^{me2a}$, and H3R2$^{me2s}$ peptides (Liu et al, 2012), the function of SMN-H3 association remains elusive. FMRP, a SMN partner, was shown to associate, via its TUDOR domain, with methylated histones in response to DNA damage (Alpatov et al, 2014). We contemplate that SMN-H3K79$^{me1}$ may potentially be involved in the regulation of gene expression, transcription, co-transcriptional splicing, or generally regulates access to genetic information.

SMN$_{TUDOR}$ is herein identified as the first reader of the H3K79$^{me1}$ mark, which is found on exons and linked to alternative splicing events. Importantly, SMA-linked SMN$_{TUDOR}$ mutants (SMN$_{ST}$) impact profound interactions with histones. Because the TUDOR royal family has always been promiscuous, it is interesting to discover that SMN$_{TUDOR}$ is having an affair with both methylated arginine and methylated lysine residues.

# Materials and Methods

### Recombinant protein expression

The cDNA of human full-length SMN and truncations were inserted in pGEX-6P-1 (GE Healthcare) using *BamHI* and *XhoI*. Aromatic cage mutants were generated by site-directed mutagenesis using Pfu Turbo (Stratagene) followed by *DpnI* (NEB) digestion. Constructs were sequence-verified (GATC Biotech AG or Biofidal) and transformed into BL21 DE3 cells (Stratagene). BL21 cells were grown overnight with ampicillin selection at 37°C with agitation. The following day, cultures were scaled up in 250 ml LB (Sigma-Aldrich) and grown at 37°C until OD$_{600}$ ~0.6. Then, expression of recombinant GST proteins was induced with 0.2 mM IPTG for 2.5–3 h at 37°C. Cells were harvested by centrifugation and resuspended in lysis buffer (50 mM Tris, pH 7.5, 150 mM NaCl, 0.05% NP-40, supplemented Complete EDTA-free [Roche]). After a brief sonication, lysates were cleared by centrifugation and incubated with glutathione–sepharose (GE Healthcare) at 4°C on a tumbler wheel. After extensive washing, GST proteins were eluted with 10 mM reduced glutathione (Sigma-Aldrich) in 50 mM Tris, pH 8.0.

### Antibodies

The α-H3 (ab1791), α-H4K20me2 (ab9052), and HRP-conjugated α-GST (ab3416) antibodies were obtained from Abcam. The α-H2A

(07-146), α-H2B (07-371), and α-H4 (62-141-13) were obtained from Millipore. Methyl-specific H3K79me1 (pAb-082), H3K79me2 (pAb-051), and H3K79me3 (pAb-068) antibodies were purchased from Diagenode. The α-H2AZ antibody was described elsewhere (Binda et al, 2013).

### Histone interactions

GST pulldowns were performed with 25 µg calf thymus histones (Worthington) and ~1 µg recombinant GST or GST-SMN in freshly made 25 mM bis-tris propane buffer (B6755; Sigma-Aldrich), pH 6.8, with 1 M NaCl and 0.05% NP-40. Glutathione–sepharose beads (GE17-5132-01) were added for an hour, then washed four times with 1 ml bis-tris-propane buffer.

### Peptide pulldowns

Peptide pulldowns were performed with 1 µg biotinylated H3 peptides (Sabra et al, 2013) and ~1 µg recombinant GST or GST-SMN in freshly prepared 25 mM bis-tris-propane buffer with 200 mM NaCl and 0.05% NP-40. Streptavidin–sepharose beads (GE17-5113-01) were added for an hour, then washed four times with 1 ml bis-tris-propane buffer.

### H3K79 recombinant nucleosomes

Bacterial expression vectors for histones H2A and H2B were purchased from Addgene (42,634 and 42,630, respectively). Plasmids to express *X. laevis* H3 (xH3) in pET-3d and xH4 in pET-3a were obtained from Professor Arrowsmith (University of Toronto). Introduction of C110A and K79C mutations in xH3 was performed by site-directed mutagenesis using QuikChange (Stratagene), and the plasmid was sequence-verified. Recombinant histones were purified from *E. coli*, and modified where indicated, before octamer assembly and subsequent refolding of rNCP with a 151 base pair 601 Widom DNA as previously described (Dyer et al, 2003; Galloy et al, 2021). Briefly, the histones were purified from inclusion bodies under denaturing conditions on a 5-ml HiTrap SP FF (GE Healthcare) cation exchange column on a next-generation chromatography (NGC, Bio-Rad). Fractions containing the purified histone were pooled and dialyzed three times into 4 liters of water and 2 mM β-mercaptoethanol before lyophilization. The four histones were then unfolded into 20 mM Tris, pH 7.5, 7 M guanidine–HCl, and 10 mM DTT and mixed in equimolar ratios before octamer refolding into 2 M NaCl, 10 mM Tris, pH 7.5, 1 mM EDTA. Octamers were then purified on a Superdex 200 HiLoad 16/600 size exclusion column (GE Healthcare) and wrapped with the 151 base pair 601 Widom DNA to obtain rNCPs. Native gel analysis was used to validate the quality of the reconstitution.

### Histone labeling

The installation of a monomethyl–lysine analog at the mutated cysteine of the H3K_C79 (C110A) histone was carried out as described (Simon et al, 2007) using the 2-chloro-N-methylethanamine hydrochloride (Toronto Research Chemicals C428323) to generate H3K_C79me1 (C110A). The installation of the analog was confirmed by electrospray ionization mass spectrometry on a LC-ESI-QTOF Agilent 6538 mass spectrometer and immunoblotting against H3K79me1 (Fig S2).

## Supplementary Information

## Acknowledgements

We thank Dr Faouzi Baklouti (DR2 Inserm) for proofreading and helpful discussions. This work was supported by AFM Téléthon grants awarded to P Lomonte (Plans stratégiques MyoNeurALP and MyoNeurALP2) and funding from the Joint Collaborative Research Program of University of Ottawa Centre for Neuromuscular Disease and Claude Bernard Université Lyon 1 Institute NeuroMyoGene awarded to O Binda and P Lomonte. J Côté is funded by a Canadian Institutes of Health Research (CIHR) grant (MOP 123381) and CureSMA Canada. A Fradet-Turcotte is funded by the Natural Sciences and Engineering Research Council of Canada (NSERC), grant (RGPIN-2016-05844). A Fradet-Turcotte is a tier 2 Canada Research Chair in Molecular Virology and Genomic Instability and is supported by the Foundation J.-Louis Lévesque. M Galloy is supported by a doctoral fellowship from the Fonds de Recherche du Québec - Santé (FRQS). P Lomonte is a CNRS research director.

### Author Contributions

O Binda: conceptualization, supervision, funding acquisition, investigation, methodology, and writing—original draft, review, and editing.
AB Kimenyi Ishimwe, M Galloy, and K Jacquet: investigation.
A Corpet: conceptualization and supervision.
A Fradet-Turcotte: supervision, funding acquisition, and writing—original draft.
J Côté: conceptualization, supervision, and funding acquisition.
P Lomonte: conceptualization, supervision, funding acquisition, writing—original draft, and project administration.

### Conflict of Interest Statement

The authors declare that they have no conflict of interest.

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
