## [Reviewer comments · Life Science Alliance]

Life Science Alliance

The TUDOR domain of SMN is an H3K79me1 histone mark reader

Olivier Binda, Aimé Boris Kimenyi Ishimwe, Maxime Galloy, Karine Jacquet, Armelle Corpet, Amélie Fradet-Turcotte, Jocelyn Côté, and Patrick Lomonte

DOI: <https://doi.org/10.26508/lsa.202201752>

Corresponding author(s): *Olivier Binda, University of Ottawa*

Review Timeline:

Submission Date:	2022-10-05
Editorial Decision:	2022-12-01
Revision Received:	2023-02-14
Editorial Decision:	2023-02-28
Revision Received:	2023-02-28
Accepted:	2023-03-01

Scientific Editor: Novella Guidi

Transaction Report:

December 1, 2022

Re: Life Science Alliance manuscript #LSA-2022-01752

Dr. Olivier Binda
University of Ottawa
Department of Cellular and Molecular Medicine
451 Smyth Road
Ottawa, Ontario K1H 8M5
Canada

Dear Dr. Binda,

Thank you for submitting your manuscript entitled "The TUDOR domain of SMN is an H3K79me1 histone mark reader" to Life Science Alliance. The manuscript was assessed by expert reviewers, whose comments are appended to this letter. We invite you to submit a revised manuscript addressing the Reviewer comments.

Thank you for this interesting contribution to Life Science Alliance. We are looking forward to receiving your revised manuscript.

Sincerely,

B. MANUSCRIPT ORGANIZATION AND FORMATTING:

Reviewer #1 (Comments to the Authors (Required)):

In this interesting manuscript, Binda and colleagues follow up on their previous cell biology work suggesting that the survival motor neuron (SMN) protein, also called GEMIN1, interacts with histone H3 monomethylated at lysine 79 (H3K79me1) in cells in that SMN localizes with damaged centromeres in a DOT1L methyltransferase-dependent manner. DOT1L generates H3K79me1. Using biochemical approaches, the authors now present convincing evidence that SMN specifically binds H3K79me1 via a single TUDOR domain. This is notable because the TUDOR domain of SMN had previously been shown to bind arginine-methylated proteins. And so, this TUDOR domain has a dual function. The pull-downs in Figure 5B show markedly stronger signals for SMN binding to H3K79me2 than for 53BP1 binding to H3K79me2. In this assay, 53BP1 could be seen as a "negative control" rather than a "positive control" (the authors may comment). This is because 53BP1 has been shown to bind H3K79me2 with very low affinity (millimolar Kd), which likely reflects the preference of 53BP1 for a demethylated lysine (me2) over other methylation states (me1 or me3) regardless of the amino acid sequence surrounding me2. Indeed, 53BP1 also binds other sites in other dimethylated histone peptides with similarly low affinity as shown by Tong et al. (Structure. 2015 Feb 3;23(2):312-21. doi: 10.1016/j.str.2014.11.013.). Higher affinity binding of 53BP1 (micromolar Kd) has only been shown for p53K382me2, H4K20me2 and H3K36me2 peptides. Further supporting the specific interaction of SMN with methylated H3K79, in Figure 5D, SMN binds H3K79me1 markedly better than 53BP1 binds H3K79me2. This figure panel also highlights the preference of SMN for a mono-methyllysine (at the H3K79 site). Ultimately, the most convincing results are in Figure 6, where SMN is shown to preferentially bind H3K79me1 in the nucleosome. In summary, this paper convincingly shows that SMN specifically binds H3K79me1 via its TUDOR domain. This is an important result because deletions or mutations in SMN have been linked to spinal motor atrophy (SMA) and several of the mutations are in the TUDOR domain. Several relevant mutations in SMN TUDOR from SMA patients were tested in the manuscript and shown to prevent binding to the H3K79me1 nucleosome. The work is technically sound and is suitable for publication in LSA.

Reviewer #2 (Comments to the Authors (Required)):

Spinal Muscular Atrophy (SMA) is an autosomal recessive childhood motor neuron disorder characterised by the degeneration of the lower motor neurons in the anterior horn of the spinal cord. It has been well known that depletion of the survival motor neuron (SMN) protein leads to SMA disease. The manuscript follows up a previous study (Sabra et al., 2013) to further characterise the interaction between SMN and histone H3 methylated on lysine 79. The authors use biochemical assays and a series of SMN mutants to show that the Tudor domain of the SMN protein associates directly in vitro with H3K79me1. The current format of the manuscript herein raises several questions and concerns that need to be addressed. The manuscript would be significantly improved following the comments below:

Major comments:

1. The authors refer to the SMN protein being 'lost' or 'deleted' rather than 'depleted'; this needs to be amended across the entire manuscript when referring to SMN protein levels.
2. The authors need to re-write the 'summary blurb'. This section needs to be informative and describe the context and significance of the results obtained.
3. The introduction is too brief and a better overview of the SMA field is needed. For example, SMN has also other functions (apart from snRNP assembly). In addition, information provided for treatment is not accurate. The introduction states the first gene therapy to be approved was in 2019 - but the first treatment approved was Spinraza in 2016. While Spinraza is not considered "gene therapy" by the FDA, it should be mentioned that there are currently two more (2) treatments approved for SMA patients. Furthermore, the authors refer to their previous publication (Sabra et al., 2013) but no explanation is given about DOT1L; need to clarify to the reader what this is. Moreover, the authors mention two SMN genes and state that loss of SMN1 results in low levels of SMN but do not explain why this is - it would be important to the reader to explain the role of SMN2 (including its role as a protective modifier).
4. The authors need to blot both the lysates and the immunoprecipitates for all the conditions with all the antibodies used. This applies to all the relevant figures (Figures 1-6 and supplementary ones).
5. Figure legends need to be explained clearly and thoroughly, providing the reader with all the information necessary to understand each figure without returning to the main manuscript; all figure legends need to be re-visited. Authors should also avoid 'as in panel X etc'.
6. The authors state that the full-length form of SMN failed to interact with H3 but this is not supported by Figure 2B. Authors

need to check that all observations listed on the results' section are supported by the corresponding figures presented.

7. 'These results suggest that SMA may arise in some cases from impaired SMN-H3 interactions or other interactions requiring an intact SMNTUDOR.'; there is no functional data to support this; statement needs to be rephrased.

8. Discussion is also relatively small. What other studies exist in relation to SMN and histone modifications? What is the bigger picture here?

Other minor comments:

1. Abstract: 'histone marker reader' repeated twice in the same sentence; needs re-phrasing.

2. Abstract: 'mutational analyzes' need to be amended to 'mutational analysis'.

3. The authors should avoid using the abbreviation SMNst

4. Introduction: '..monogenic pathology of SMA'; needs to be added.

5. Full names need to be provided across the text before abbreviations given (e.g. snRNPs etc); these need to be amended across the manuscript.

6. The authors use the symbol ' Δ ' in the text but 'D' for the Figures; consistency between manuscript text and manuscript figures is important.

7. '..if not better', '..so far..'; authors should avoid such phrases

Reviewer #3 (Comments to the Authors (Required)):

In this manuscript, Binda et al., use overexpressing constructs to determine that the functional TUDOR domain in the survival motor neuron (SMN) protein specifically binds to histone H3 monomethylated on lysine 79 (H3K79me1). SMN is a ubiquitously expressed protein, which when deleted and/or mutated leads to the neuromuscular condition spinal muscular atrophy (SMA). The authors also show that certain SMA-relevant TUDOR mutations lose their ability to bind to H379me1. Overall, this manuscript provides for the first time evidence of an interaction between the SMN TUDOR domain and H379me1.

Comments/Suggestions:

1) Overall, the manuscript contains very field-specific expert terminology throughout (e.g. reader in the title), which may limit accessibility to a wider range of researchers. Perhaps it would be beneficial for the authors to provide brief definitions and meaning for expert terminologies used throughout the manuscript to increase reach and impact.

2) SMA should be written in full when first mentioned in the Introduction.

3) In the first paragraph of the Introduction, it is not clear to a non-SMA expert which therapy is being referred to and to an SMA expert, why only one therapy was mentioned out of the three therapeutics. In fact, it is this reviewer's opinion that the first paragraph about therapies is not needed. It is ok to investigate SMN function just for the sake of it, particularly that therapies are not mentioned elsewhere in the manuscript.

4) In the second paragraph of the Introduction, the authors mention the SMA was found to phase separate. What is the significance of this?

5) Is the differential binding of H3 by SMA-relevant TUDOR mutants associated with severity of disease?

6) This paper relies on exogenous constructs outside of more natural environments. Have the authors considered verifying these findings in patient fibroblasts for example?

7) In the Discussion and Conclusion, please could the authors provide more insights on the significance of the SMN-H3K79me1 interaction? What does it do? And what role does it play perhaps in SMA?

8) For each figure, could more information be provided on how many times the pull-down experiments were performed to confirm the findings presented as representative images.

9) Please provide stats directly on Figure 3C.

RESPONSE TO REVIEWERS COMMENTS

Reviewer #1

In this interesting manuscript, Binda and colleagues follow up on their previous cell biology work suggesting that the survival motor neuron (SMN) protein, also called GEMIN1, interacts with histone H3 monomethylated at lysine 79 (H3K79me1) in cells in that SMN localizes with damaged centromeres in a DOT1L methyltransferase-dependent manner. DOT1L generates H3K79me1. Using biochemical approaches, the authors now present convincing evidence that SMN specifically binds H3K79me1 via a single TUDOR domain. This is notable because the TUDOR domain of SMN had previously been shown to bind arginine-methylated proteins. And so, this TUDOR domain has a dual function. The pull-downs in Figure 5B show markedly stronger signals for SMN binding to H3K79me2 than for 53BP1 binding to H3K79me2. In this assay, 53BP1 could be seen as a "negative control" rather than a "positive control" (the authors may comment). This is because 53BP1 has been shown to bind H3K79me2 with very low affinity (millimolar Kd), which likely reflects the preference of 53BP1 for a demethylated lysine (me2) over other methylation states (me1 or me3) regardless of the amino acid sequence surrounding me2. Indeed, 53BP1 also binds other sites in other dimethylated histone peptides with similarly low affinity as shown by Tong et al. (Structure. 2015 Feb 3;23(2):312-21. doi: 10.1016/j.str.2014.11.013.). Higher affinity binding of 53BP1 (micromolar Kd) has only been shown for p53K382me2, H4K20me2, and H3K36me2 peptides. Further supporting the specific interaction of SMN with methylated H3K79, in Figure 5D, SMN binds H3K79me1 markedly better than 53BP1 binds H3K79me2. This figure panel also highlights the preference of SMN for a mono-methyllysine (at the H3K79 site). Ultimately, the most convincing results are in Figure 6, where SMN is shown to preferentially bind H3K79me1 in the nucleosome. In summary, this paper convincingly shows that SMN specifically binds H3K79me1 via its TUDOR domain. This is an important result because deletions or mutations in SMN have been linked to spinal motor atrophy (SMA) and several of the mutations are in the TUDOR domain. Several relevant mutations in SMN TUDOR from SMA patients were tested in the manuscript and shown to prevent binding to the H3K79me1 nucleosome. The work is technically sound and is suitable for publication in LSA.

Thank you very much for the highly positive comments.

We called 53BP1 a positive control since it was shown to interact with H3K79me2, although very weakly. It is not the current view that 53BP1 associates with H3K79^{me2}, but with H4K20^{me2}, with higher affinity. However, 53BP1 is still reported as an H3K79^{me2} reader and we do observe some interaction between 53BP1 and H3K79^{me2} peptide and H3 from calf thymus histone mixture (Figure 5).

Reviewer #2

Spinal Muscular Atrophy (SMA) is an autosomal recessive childhood motor neuron disorder characterised by the degeneration of the lower motor neurons in the anterior horn of the spinal cord. It has been well known that depletion of the survival motor neuron (SMN) protein leads to SMA disease. The manuscript follows up a previous study (Sabra et al., 2013) to further characterise the interaction between SMN and histone H3 methylated on lysine 79. The authors use biochemical assays and a series of SMN mutants to show that the Tudor domain of the SMN protein associates directly in vitro with H3K79me1. The current format of the manuscript herein raises several questions and concerns that need to be addressed. The manuscript would be significantly improved following the comments below:

Major comments:

1. The authors refer to the SMN protein being 'lost' or 'deleted' rather than 'depleted'; this needs to be amended across the entire manuscript when referring to SMN protein levels.

We agree with the reviewer. Indeed, *SMN1* the gene is deleted or mutated in SMA patients, however, *SMN2* remains (often with more than 1 copy) and able to produce some level of SMN protein. We corrected "lost" by "depleted levels" in the abstract (the term "lost" is not used anywhere else).

2. The authors need to re-write the 'summary blurb'. This section needs to be informative and describe the context and significance of the results obtained.

The Summary Blurb length is limited to 200 characters (including spaces) and thus very concise. A new shorter Summary Blurb was written:

The Survival of Motor Neuron protein is depleted in Spinal Muscular Atrophy pathology and herein defined as the first reader of histone H3 mono-methylated on lysine 79 through its central TUDOR domain.

3. The introduction is too brief and a better overview of the SMA field is needed. For example, SMN has also other functions (apart from snRNP assembly). In addition, information provided for treatment is not accurate. The introduction states the first gene therapy to be approved was in 2019 - but the first treatment approved was Spinraza in 2016. While Spinraza is not considered "gene therapy" by the FDA, it should be mentioned that there are currently two more (2) treatments approved for SMA patients. Furthermore, the authors refer to their previous publication (Sabra et al., 2013) but no explanation is given about DOT1L; need to clarify to the reader what this is. Moreover, the authors mention two SMN genes and state that loss of SMN1 results in low levels of SMN but do not explain why this is - it would be important to the reader to explain the role of SMN2 (including its role as a protective modifier).

The Introduction has been lengthened according to the reviewer's suggestions, including additional information on DOT1, clarifications on the differences between *SMN1* and *SMN2*. However, we didn't expand the therapy section, in agreement with Reviewer 3 and because the article remains very biochemical in essence and not clinical at all.

4. The authors need to blot both the lysates and the immunoprecipitates for all the conditions with all the antibodies used. This applies to all the relevant figures (Figures 1-6 and supplementary ones).

Figure 6 has been modified to include the Ponceau staining to indicate the levels of GST-tagged proteins pulled down.

5. Figure legends need to be explained clearly and thoroughly, providing the reader with all the information necessary to understand each figure without returning to the main manuscript; all figure legends need to be re-visited. Authors should also avoid 'as in panel X etc'.

More details are now provided. However, some "as in panel" usage was retained to avoid unnecessary redundancy.

6. The authors state that the full-length form of SMN failed to interact with H3 but this is not supported by Figure 2B. Authors need to check that all observations listed on the results' section are supported by the corresponding figures presented.

In describing Figure 2B we state that SMN_{TUDOR} fails to associate with H3 when compared with full-length SMN (i.e. the TUDOR domain on its own (SMN_{TUDOR}) failed to associate with H3 as well as the full-length form of SMN). This is now clarified ("as well as" was replaced by "when compared with").

7. 'These results suggest that SMA may arise in some cases from impaired SMN-H3 interactions or other interactions requiring an intact SMN_{TUDOR} .'; there is no functional data to support this; statement needs to be rephrased.

Well, we say "suggest" and use "may", which means hypothetically that maybe there is something happening... something that Reviewer 3 (point 7) asked us to elaborate on, discuss, hypothesize....

In fact, SMN_{ST} mutants do indeed cause SMA, they were identified in SMA patients that retain *SMN1*.

8. Discussion is also relatively small. What other studies exist in relation to SMN and histone modifications? What is the bigger picture here?

As far as we know, there are no other studies regarding SMN-H3 interactions. There is however some work regarding $FMRP_{TUDOR}$ interactions with H3 in the context of DNA damage. We have expanded the discussion section accordingly.

Other minor comments:

1. Abstract: 'histone marker reader' repeated twice in the same sentence; needs re-phrasing.

Replaced the first one by protein known to associate with H3K79^{me1}.

2. Abstract: 'mutational analyzes' need to be amended to 'mutational analysis'.

Different mutants were used in a number of assays justifying the term analyzes (plural) instead of analysis (singular).

3. The authors should avoid using the abbreviation SMNst.

It is short for SMA-linked TUDOR domain mutations in SMN. We find the abbreviation SMN_{ST} quite convenient in writing, but also in reading.

4. Introduction: "...monogenic pathology of SMA"; needs to be added.

It is in the 2nd line of the introduction.

5. Full names need to be provided across the text before abbreviations given (e.g. snRNPs etc); these need to be amended across the manuscript.

Small nuclear ribonucleoproteines is now spelled out in the introduction and other terms spelled out (e.g. RNA polymerase II, glutathione S-transferase).

6. The authors use the symbol 'Δ' in the text but 'D' for the Figures; consistency between manuscript text and manuscript figures is important.

The Δ symbol was likely converted to a capital D during handling of the original files by journal staff. The original files all contain Δ (or other) symbol. We'll need to ask LSA to correct this.

7. '..if not better', '..so far..'; authors should avoid such phrases.

Deleted "if not better". The first instance of "so far" is deleted and the second replaced by "until now".

Reviewer #3

In this manuscript, Binda et al., use overexpressing constructs to determine that the functional TUDOR domain in the survival motor neuron (SMN) protein specifically binds to histone H3 monomethylated on lysine 79 (H3K79me1). SMN is a ubiquitously expressed protein, which when deleted and/or mutated leads to the neuromuscular condition spinal muscular atrophy (SMA). The authors also show that certain SMA-relevant TUDOR mutations lose their ability to bind to H379me1. Overall, this manuscript provides for the first time evidence of an interaction between the SMN TUDOR domain and H379me1.

Overall we appreciate the reviewer's comments. To clarify, we did not over express any proteins, the work was accomplished *in vitro* with purified recombinant proteins.

Comments/Suggestions:

1) Overall, the manuscript contains very field-specific expert terminology throughout (e.g. reader in the title)), which may limit accessibility to a wider range of researchers. Perhaps it would be beneficial for the authors to provide brief definitions and meaning for expert terminologies used throughout the manuscript to increase reach and impact.

We would welcome suggestions to replace the term reader by something else, but then it is the name for that category of proteins and is broadly used in the literature.

2) SMA should be written in full when first mentioned in the Introduction.

Done. It is also spelled out in the keywords section and the abstract.

3) In the first paragraph of the Introduction, it is not clear to a non-SMA expert which therapy is being referred to and to an SMA expert, why only one therapy was mentioned out of the three therapies. In

fact, it is this reviewer's opinion that the first paragraph about therapies is not needed. It is ok to investigate SMN function just for the sake of it, particularly that therapies are not mentioned elsewhere in the manuscript.

Reviewer 2 wants more information on therapies, but we agree that therapies need not be mentioned. See amended first paragraph.

4) In the second paragraph of the Introduction, the authors mention the SMA was found to phase separate. What is the significance of this?

We don't know yet. However, SMN mutations found in SMA patients fail to phase separate and also fail to associate with H3K79^{me1}. Could phase separation enable association with H3K79^{me1}?

5) Is the differential binding of H3 by SMA-relevant TUDOR mutants associated with severity of disease?

To address this point, we have now included the type of SMA in the summary Table with the binding of SMN to H3 and nucleosomes.

6) This paper relies on exogenous constructs outside of more natural environments. Have the authors considered verifying these findings in patient fibroblasts for example?

The work is not exogenous, it is biochemical, with no otherwise pretentions. Previous work looked at SMN-H3 interactions by chromatin immunoprecipitation (ChIP) (Sabra *et al* 2013). It would be interesting to investigate the location of SMN and H3K79^{me1} genome-wide by ChIP-seq, but this is beyond the scope of this study.

7) In the Discussion and Conclusion, please could the authors provide more insights on the significance of the SMN-H3K79me1 interaction? What does it do? And what role does it play perhaps in SMA?

We have now elaborated on this point in the discussion and mention regulation of gene expression, transcription, splicing co-transcriptionally, access to genetic information as potential functions for SMN-H3K79^{me1}.

8) For each figure, could more information be provided on how many times the pull-down experiments were performed to confirm the findings presented as representative images.

It is now specified.

9) Please provide stats directly on Figure 3C.

We've indicated the p-values that were significant (SMN vs W102A/Y130A and SMN vs W102A/Y109A/Y130A).

February 28, 2023

RE: Life Science Alliance Manuscript #LSA-2022-01752R

Dr. Olivier Binda
University of Ottawa
Department of Cellular and Molecular Medicine
451 Smyth Road
Ottawa, Ontario K1H 8M5
Canada

Dear Dr. Binda,

Thank you for submitting your revised manuscript entitled "The TUDOR domain of SMN is an H3K79me1 histone mark reader". We would be happy to publish your paper in Life Science Alliance pending final revisions necessary to meet our formatting guidelines.

- please add the author contributions and a conflict of interest statement to the main manuscript text
- please incorporate your conclusions section in the discussion section

Figure Check:

- please add weights next to blots
- Figure 5C middle row: shading after 1st blot is strange. Please provide source data for this panel

A. FINAL FILES:

B. MANUSCRIPT ORGANIZATION AND FORMATTING:

Sincerely,

Reviewer #2 (Comments to the Authors (Required)):

I am happy with the revisions made.
No further changes needed from my part.

Reviewer #3 (Comments to the Authors (Required)):

The authors have addressed all of my initial comments. And apologies for incorrectly stating that the study included exogenous/overexpression experiments.

March 1, 2023

RE: Life Science Alliance Manuscript #LSA-2022-01752RR

Dr. Olivier Binda
University of Ottawa
Department of Cellular and Molecular Medicine
451 Smyth Road
Ottawa, Ontario K1H 8M5
Canada

Dear Dr. Binda,

Thank you for submitting your Research Article entitled "The TUDOR domain of SMN is an H3K79me1 histone mark reader". It is a pleasure to let you know that your manuscript is now accepted for publication in Life Science Alliance. Congratulations on this interesting work.

DISTRIBUTION OF MATERIALS:

Again, congratulations on a very nice paper. I hope you found the review process to be constructive and are pleased with how the manuscript was handled editorially. We look forward to future exciting submissions from your lab.

Sincerely,
